# Factors Related to Oral-Health-Related Quality of Life in Adult Male Inmates: A Cross-Sectional Study

**DOI:** 10.3390/healthcare11212848

**Published:** 2023-10-29

**Authors:** Hae-Soo Yoon, Kyoung-Seon Kim, Jong-Hwa Jang

**Affiliations:** 1Department of Public Health Science, Dankook University Graduate School, Cheonan-si 31116, Chungcheongnam-do, Republic of Korea; foxyyhhs@gmail.com; 2Department of Dental Hygiene, Jeonju Kijeon College, Jeonjucheonseo-ro, Jeonju-si 54989, Jeollabuk-do, Republic of Korea; kk2212@hanmail.net; 3Department of Dental Hygiene, College of Health Science, Dankook University, Cheonan-si 31116, Chungcheongnam-do, Republic of Korea

**Keywords:** inmate, oral health behaviour, oral-health-related quality of life, oral symptom awareness, toothbrushing

## Abstract

The level of oral health among inmates living in restricted environments is poor. We investigated oral health behaviours and oral symptoms among male inmates and analysed factors related to OHRQoL. Three hundred and seventy-six male inmates in two prisons were investigated using a self-reported questionnaire. The survey items included oral health behaviours, oral symptoms, self-esteem, and OHRQoL level. We analysed the cross-correlations and factors related to OHRQoL. Toothbrushing after lunch and dinner was practiced by 83.5% of male inmates. ‘Food impaction’ was experienced by 45.4%. The OHRQoL level among male inmates was low, at 39.90 points. A hierarchical multiple regression model that controlled for general characteristics identified the following OHRQoL factors in order of significance: trouble biting/chewing (β = −0.307), toothache (β = −0.154), temporomandibular joint disorder (β = −0.099), and periodontal health (β = 0.089). Self-esteem and OHRQoL were not significantly correlated. To ensure oral health in male inmates, the results of this study suggest that an oral health education programme, as well as the provision of adequate dental care services, is required to increase appropriate oral health practice rates.

## 1. Introduction

Since prison inmates are members of the public who are set to eventually return to society, healthcare management is crucial in maintaining their physical and mental health [1]. Inmates in Korea are more likely to have poor health due to the significant healthcare gap compared to even the low-income group of the general population [2]. According to a report based on a survey of the right to health in detention facilities, ‘dental consultation and treatment’ ranked first in terms of the services considered most necessary in relation to health in the current Korean prison system, followed by ‘periodic health examination’ and the ‘expansion of the use of external hospital treatment’ [3]. In addition, the prevalence of major diseases was shown to be high among those with dental diseases (6.3%), along with hypertension (14.4%) and diabetes (8.5%); in particular, tooth diseases (4.6%) and gum diseases (2.2%) accounted for the majority [3].

The oral cavity plays a vital role as a link between the external environment and the body [4,5]. Poor oral hygiene causes various oral diseases, such as dental caries, periodontal disease, and tooth loss, which significantly affect oral health [6,7]. Considering this relationship, oral health behaviours and related factors should be investigated among inmates to resolve their health problems.

The oral health status and oral hygiene habits affect oral-health-related quality of life (OHRQoL) [8]. To evaluate OHRQoL, the Oral Health Impact Profile 14 (OHIP-14) questionnaire is typically used. This questionnaire comprises concepts such as functional limitations, physical pain, psychological discomfort, physical disabilities, and social handicaps. Factors affecting the OHIP-14 have been reported to include not only sociodemographic characteristics, but also sociopsychological factors, such as the oral health status and self-esteem [9,10,11].

A recent study from India reported a high level of oral disease among inmates; additionally, the length of imprisonment was shown to affect the degree of dental caries experienced [12]. In total, 33% of female inmates in Brazil experienced tooth loss after imprisonment, and 65.6% subjectively perceived that prison dental services were of below-average quality, which negatively affected their quality of life [13]. Deterioration in inmates’ quality of life was associated with caries, dental pain, deep periodontal pockets, prosthesis use, and age [14]. Moreover, adult male inmates had worse oral health than did the general adult male population, with 7.37 times more oral problems affecting their daily life [15]. Among imprisoned male adolescents, the tooth-loss rate was 47.06% [16], the prevalence of dental caries was high, and oral hygiene conditions were poor, requiring special attention from a policy perspective [17]. On the other hand, interventions for improving oral health among inmates have been reported to have a positive effect on oral-health-related behavioural changes [18]. According to a review article on oral health among inmates in Europe, oral health among this population should be understood from an organisational perspective due to its complexity, and more studies are needed to address gaps in the literature on oral health in inmates [19].

A comparison of the prevalence of dental caries among detention centre inmates based on the Korea National Health and Nutrition Examination Survey (KNHANES), as a study on the oral health among detention centre inmates in Korea, showed that the oral health among the inmate group was worse than that among the general population [20]. In addition, conservative infection control in a case of acute mandibular pericoronitis in an inmate [21] and the oral health status and awareness of oral health among adolescents in juvenile protection facilities and detention centre inmates have been reported [1,22,23]. Nevertheless, currently, insufficient studies have investigated the factors related to oral health in adult inmates. Considering the results of several previous studies, oral health behaviours, perceived oral symptoms, and self-esteem among inmates are predicted to be significantly related to OHRQoL [11,13,14,15].

This study investigated oral health behaviours, perceived oral symptoms, and OHRQoL levels among inmates in Korean prisons, and sought to identify cross-correlations and factors affecting OHRQoL to be utilised as a basis for developing programmes to improve oral health in inmates.

## 2. Materials and Methods

### 2.1. Study Design

This cross-sectional, correlational survey study investigated oral health behaviours and perceived oral symptoms among male inmates and sought to identify factors associated with OHRQoL. The study adhered to the tenets of the Declaration of Helsinki and was approved by the institutional review board of Dankook University (IRB: DKU IRB 2021-08-029). Data were registered with the Clinical Research Information Service in Korea (CRIS) (KCT 0008129) [24]. Participant consent was obtained before conducting the survey.

### 2.2. Participants

The study participants were male inmates from two prisons in the Chungcheong region of Korea. The appropriate sample size was calculated using G-power 3.1.7 (https://download.cnet.com/G-Power/3000-2054_4-10647044.html (accessed on 1 September 2021)). With the effect size set at 0.10, power at 0.90, and 27 predictive factors using a multiple regression analysis, we calculated that 313 participants were required. By estimating a dropout rate due to incomplete survey responses of approximately 20%, we determined that approximately 376 participants were required.

The participants selected were inmates who voluntarily agreed to respond to the questionnaire.

### 2.3. Variables

The questionnaires included items on sociodemographic characteristics (age), oral health behaviours (2 items for oral health awareness, 2 items for the use of dental treatment, and 3 items for oral health care), OHRQoL (14 items), self-esteem (10 items), and perceived oral symptoms (8 items).

#### 2.3.1. Dependent Variables

For the OHRQoL scale, the OHIP-14 scale, reported by Slade and Spencer in 1994, was used [19,25]. The OHIP-14 scale was scored as follows: very often, 0 points; fairly often, 1 point; occasionally, 2 points; hardly ever, 3 points; and never, 4 points. The sum was calculated using inverse transformation processing. The highest possible score was 56 points. The higher the score, the higher the OHRQoL was. The internal consistency of this tool, as assessed with Cronbach’s alpha, was 0.88.

#### 2.3.2. Independent Variables

As for the perceived oral symptom scale, eight items included in the 2021 Youth Health Risk Behaviour Web-Based Survey were used [26]. These items were as follows: Q1. Does food get stuck between your teeth? Q2. Are you sensitive to hot or cold food? Q3. Have you ever experienced bleeding gums or gum disease? Q4. Do you feel like you have an unpleasant bad breath (oral malodour) in your mouth? Q5. Is your mouth often dry? Q6. Do you have trouble chewing or biting food? Q7. Have you ever felt that your jaw is dislocated or hurts when you open your mouth wide or chew? Q8. Have you ever had tooth pain (toothache) even when your mouth is still? Perceived oral symptoms were measured on a 5-point Likert scale as follows: not at all, 1 point; very rarely, 2 points; sometimes, 3 points; often, 4 points; and always, 5 points. The mean score of the eight items was calculated. The highest possible score was 5 points. The higher the score, the more severe the perceived oral symptoms were. The internal consistency of this tool, as calculated with Cronbach’s alpha, was 0.83.

For the self-esteem scale, the tool developed by Rosenberg (1965) was used [27,28]. Each item was scored on a 5-point Likert scale as follows: strongly disagree, 1 point; disagree, 2 points; neutral, 3 points; agree, 4 points; and strongly agree, 5 points. The mean of the scores of the 10 items was calculated. The highest possible score was 5 points. The higher the score, the higher the individual’s level of self-esteem was. The internal consistency of this tool, as assessed using Cronbach’s alpha, was 0.92.

#### 2.3.3. Covariables

Age, a sociodemographic characteristic, was investigated as a continuous variable, and was then recoded as a categorical variable (aged between 20 and 29 years = 1; aged between 30 and 39 years = 2; aged between 40 and 49 years = 3; aged between 50 and 59 years = 4; aged 60 years or older = 5).

Oral health behaviour was investigated with multiple responses for the timing of brushing teeth during the previous day (before breakfast = 1; after breakfast = 2; before lunch = 3; after lunch = 4; before dinner = 5; after dinner = 6; after snack time = 7; before sleeping = 8) in reference to a previous study [29]. Dental treatment experiences and dental examinations in the past year (yes = 1; no = 2; do not know = 3), toothbrushing time (less than 1 min = 1; 1 min to less than 2 min = 2; 2 min to less than 3 min = 3; more than 3 min = 4), use of fluoride toothpaste, and ability to chew food with increasing hardness levels (tofu and rice, apple, kimchi, meat, dried squid, and hard candy) were investigated as categorical items. Subjective health and oral health awareness were measured on a 5-point Likert scale (strongly agree = 5 points; to strongly disagree = 1 point). Higher scores were considered to correspond to higher subjective health and oral health awareness in participants.

### 2.4. Data Collection

From 23–30 September 2021, the researcher visited the prison in person, explained the purpose of the study through the warden of the correctional institution to obtain consent, and delivered a structured questionnaire to the correctional officers. The questionnaire was distributed to inmates who voluntarily agreed to respond to the survey during lunchtime through a study participant recruitment notice displayed at the cafeteria. The survey process was performed anonymously, and a toothbrush and toothpaste set was provided as a reward for responding to the survey. A total of 376 inmates participated. The completed questionnaires were collected by putting them in a prepared questionnaire collection box. No incomplete questionnaires were included; thus, 376 (100.0%) questionnaires were included for the analysis.

### 2.5. Statistical Analyses

SPSS software (Version 23.0; IBM Corp., Armonk, NY, USA) was used to obtain frequencies, percentages, and descriptive statistics of all the variables. Differences in the OHIP-14 according to general characteristics were analysed using an independent *t*-test or one-way analysis of variance, and Scheffe’s multiple comparison test was performed as a post hoc test. Pearson’s correlation analysis was performed to evaluate the relationship between the OHIP-14 scores and perceived oral symptoms. A hierarchical multiple regression analysis was performed to identify factors related to the OHIP-14. The significance level of this study was set at α = 0.05.

## 3. Results

### 3.1. Toothbrushing Behaviour and Perceived Oral Symptoms in Male Inmates

Among the participating male inmates, 83.5% brushed their teeth after lunch and after dinner, and 82.6% brushed their teeth after breakfast; however, only 38.0% of inmates brushed their teeth before sleeping (Figure 1). In addition, 35.5% of inmates brushed their teeth before breakfast. Oral symptoms that were perceived to occur ‘often’ or more were as follows in order: ‘food impaction (45.4%)’, ‘periodontal health (23.5%)’, ‘sensitivity to cold/hot food (21.7%)’, ‘trouble in biting/chewing (19.1%)’, and ‘malodour (15.9%)’. Among the perceived oral symptoms, the incidence of ‘TMJ disorder (8.4%)’ and ‘toothache (8.7%)’ was relatively low (Figure 2).

### 3.2. OHIP-14 According to the General Characteristics of Male Inmates

The OHIP-14 score among male inmates was low, at 39.9 ± 12.38 points. Table 1 shows the comparison of the differences according to the general characteristics of the male inmates.

The OHIP-14 scores decreased significantly with increasing inmate age (*p* < 0.001). The OHIP-14 score was higher for those in their 20s (46.86 ± 10.57) than for those in their 40s (39.84 ± 13.39), 50s (36.15 ± 12.07), and 60s or older (36.39 ± 11.28), and the scores of those in their 30s (43.60 ± 10.84) were significantly different from those in their 40s or older.

The OHIP-14 score was lower in the group that reported brushing teeth for less than 2 min than in the group who reported brushing for more than 2 min (*p* = 0.020). This score also increased as toothbrushing lengthened, in the order as follows: ‘2 to less than 3 min (41.87 ± 10.84)’, ‘more than 3 min (41.73 ± 13.12)’, and ‘1 to less than 2 min (38.10 ± 13.06)’ (*p* = 0.020).

The OHIP-14 score increased depending on chewing discomfort with harder food (*p* < 0.001). The group that could chew tofu and rice (25.81 ± 13.04) and apples (26.22 ± 15.47) had a lower OHIP-14 score than the group that could chew meat (37.30 ± 9.04), dried squid (36.45 ± 11.53), and hard candy (45.46 ± 8.80). In addition, the group that could chew kimchi (29.06 ± 12.80) had a lower OHIP-14 score than the group that could chew hard candy.

The perceived level of subjective health and oral health was found to be directly proportional to the OHIP-14 score (*p* < 0.001).

Furthermore, self-esteem was also found to increase with an increase in the OHIP-14 score in the order of ‘high’ (41.10 ± 15.64), ‘usual’ (40.12 ± 11.97), and ‘low’ (38.65 ± 10.55); however, this correlation was not statistically significant (*p* > 0.05). In addition, the OHIP-14 score did not differ significantly depending on dental treatment experience, dental examination in the past year, and use of fluoride toothpaste (*p* > 0.05).

### 3.3. Correlations between Perceived Oral Symptoms and Self-Esteem and the OHIP-14 in Male Inmates

Table 2 shows the results of the correlation analysis of perceived oral symptoms and self-esteem with the OHIP-14 scores in male inmates. Although perceived oral symptoms had a negative relationship with the OHIP-14 score, the relationship with self-esteem was not statistically significant (*p* > 0.05). The OHIP-14 score correlated negatively with perceived oral symptoms in the order of ‘trouble biting/chewing (r = −0.749)’, ‘toothache (r = −0.649)’, ‘periodontal health (r = −0.570)’, ‘sensitivity to cold/hot food (r = −0.546)’, ‘dry mouth (r = −0.523)’, ‘TMJ disorder (r = −0.488)’, and ‘food impaction (r = −0.338)’.

A strong positive correlation was found among the eight items of perceived oral symptoms. In particular, oral malodour was an oral symptom that was related to self-esteem in inmates (r = −0.114), showing that the worse the oral malodour, the lower their self-esteem was.

### 3.4. Factors Related to the OHIP-14 in Male Inmates

Table 3 shows the results of the hierarchical multiple linear regression analysis to identify factors that affected the OHIP-14 scores in male inmates. The correlation analysis of the eight items of perceived oral symptoms revealed a correlation coefficient of less than 0.8; thus, all items were used for the regression analysis. Age, subjective health awareness, and oral health awareness, which showed significant associations with the OHIP-14 score, were included as independent variables. In addition, ‘tofu and rice’ was set as the reference category for chewable food, a categorical variable, and dummy variables were used for the remaining five items. The assumptions of the regression analysis were tested and all of them were found to be satisfied. Since the Durbin–Watson statistic was 2.030, the autocorrelation of errors was deemed absent. Tolerance was less than 0.1 and the variance inflation factor value was 1.155–6.167, i.e., less than 10, indicating the absence of multicollinearity.

Model one yielded 49.9% (Adj. *R*^2^ = 0.499) explanatory power when the independent variable was adjusted for the dependent variable, the OHIP-14 score, and the regression model was found to be statistically significant (*F* = 44.754, *p* < 0.001). The size of the influence of factors on the OHIP-14 score was in the following order: subjective oral health awareness (β = 0.278), subjective health awareness (β = 0.200), and age (β = −0.099). For chewable food, hard candy (β = 0.522), meat (β = 0.270), and squid (β = 0.126) were found to be factors affecting the OHIP−14 score, as compared to tofu and rice.

Model two was analysed by adding eight items of perceived oral symptoms to model one. The model’s goodness of fit was significant (*F* = 51.631, *p <* 0.001). The adjusted explanatory power of the model was 69.8% (Adj. *R*^2^ = 0.698), indicating that the influence on the OHIP-14 increased by 19.9%, as compared with model one. The factor most significantly affecting the OHIP-14 score in male inmates was trouble biting/chewing (β = −0.307), followed by toothache (β = −0.154), age (β = −0.146), malodour (β = −0.131), subjective health (β = 0.131), TMJ disorder (β = −0.099), and periodontal health (β = −0.089). Among chewable foods in model one, meat and squid, which were factors affecting the OHIP-14 score, were not significant, and only hard candy (β = 0.175) was a significant affecting factor. In particular, a high subjective health and being able to bite candy were found to affect the OHIP-14 score positively, whereas trouble biting/chewing, toothache, age, malodour, TMJ disorder, and periodontal health were found to negatively affect the OHIP-14 scores.

## 4. Discussion

This study examined oral health behaviours and perceived subjective symptom levels in male inmates and determined factors related to OHRQoL. We found that the OHRQoL level among male inmates was low (39.90 points). In a hierarchical multiple regression model that controlled for general characteristics, we identified the following factors as affecting OHRQoL in order: trouble biting/chewing, toothache, age, subjective health, oral malodour, TMJ disorder, and periodontal health. We did not find a statistically significant correlation between self-esteem and OHRQoL.

Previous studies found that the level of oral health among prison inmates was poor as compared to that in the general population [30,31], and that oral health status affected self-esteem [32,33]. Moreover, considering the report that oral health problems were related to decreased quality of life, perceived subjective symptoms were reported to affect OHRQoL [34].

Although the mean practice rate of toothbrushing after meals among male inmates in Korea was 83.2%, the practice rate before breakfast (35.5%) was higher, and the practice rate before sleeping (38.0%) was rather low. Hence, education on appropriate oral health care is necessary. However, the practice rate of toothbrushing after lunch was 83.5%, approximately 15% higher than that obtained in the 2022 Community Health Survey (68.3%) [35]. In addition, when compared to the practice rate of toothbrushing after lunch among Korean workers aged 19 years or older (51.6%) in the sixth KNHANES (2013–2015), the rate of toothbrushing after lunch among male inmates was relatively high [36]. These findings were thought to be influenced by the laws stipulated, such as maintaining the cleanliness of inmates (Article 32 [1] of the Act on the Execution of Sentences and Treatment of Inmates) and regular exercise and bathing for health maintenance (Article 33 [1] of the Act on the Execution of Sentences and Treatment of Inmates), so that inmates can lead healthy and regular lives in an enclosed environment [37].

In terms of perceived oral symptoms, inmates responded to ‘often’ or ‘always’ having the following in order: ‘food impaction’, ‘periodontal health’, and ‘sensitivity to cold/hot food.’ However, in a study on perceived oral symptoms in older individuals, the order was ‘subjective oral health awareness’ (49.4%) and ‘chewing discomfort’ (49.2%) [38]. Nevertheless, the items included in perceived oral symptoms differed between these studies. On the other hand, all items for perceived oral symptoms and the level of OHRQoL showed a negative correlation. This finding was similar to the results of studies conducted on nonmedical hospital workers [39], industrial workers [26,40], and middle-aged people in Korea [41]. Against this background, customised oral healthcare programmes are needed in correctional facilities to alleviate the perceived oral symptoms among inmates [13].

The OHIP-14 score among the participants was rather low (39.9 out of 55 points). Except for participants in their 60s or older, the mean OHIP-14 score decreased as age increased. This finding was similar to the results of studies on industrial workers [40], middle-aged adults [42], and female inmates [14]. A toothbrushing time of ‘2 min to less than 3 mins’ was associated with the highest OHIP-14 score among our participants, similar to the results of studies on foreign workers [43] and patients visiting the dentist [44]. Since oral health care with correct toothbrushing can be related to quality of life, this result suggested that inmates should be motivated to use oral health education, and that the facilities and environment for brushing teeth should be improved [1].

Among male inmates, 246 (67.8%) responded ‘no’ to the use of a dental treatment over the past year, which was more than twice as many as those who responded ‘yes’ (110, 30.3%). In addition, 59% (214) responded ‘no’ to having an oral examination over the past year, approximately 1.5 times the percentage of those who responded ‘yes’ (145, 39.8%). These values were approximately twice as high as of those who had not used dental hospitals and clinics in the past year in a study on the relationship between health check-ups and unmet dental care needs among Korean adults (31.5%) [45]. The result of this study was somewhat high, with 57.4% of individuals responding that they had not undergone an oral examination in the last year. In addition, in a study on dental check-ups and unmet dental care needs based on the seventh KNHANES [46], 31.7% reported unmet dental care needs, while 56.5% of people did not undergo oral check-ups. Thus, the proportions of those who did not use a dental treatment and of those who did not undergo an oral examination were higher among inmates in the present study than among adults in the general Korean population.

As for the difference in the OHIP-14 score depending on the ability to chew food, the OHIP-14 scores of inmates who could chew tougher food was higher in the order of meat, dried squid, and hard candy, as compared to tofu and rice and apples. This finding was similar to the results of studies on nonmedical hospital workers [39], the KNHANES population [47], and OHRQoL in some adults [47,48]. These results suggested that better chewing ability indicated higher OHRQoL. In future studies, it is necessary to investigate the number of teeth in inmates and the use of dental prosthetics to determine the relationships with clinical indicators. Thus, the chewing-related indicator may be employed as an indicator of OHRQoL in inmates in the future [21].

In terms of the OHIP-14 score, a more positively health or oral health were subjectively perceived, with OHRQoL being the higher. This finding was similar to the results of studies on the patients of dental hospitals and clinics [44,49] and the general adult population based on the KNHANES [47]. Hence, subjective health and oral health awareness are also worth considering as indicators of oral health and quality of life among inmates in the future [20]. Furthermore, continuous customised oral health education is required for inmates to perceive their oral health positively [1].

This study showed that inmates with high self-esteem (41.10 points) had higher OHIP-14 scores than inmates with low self-esteem did (38.65 points), but the difference was not statistically significant. This finding was similar to that of previous studies in childcare teachers [50], a general adult population [51], and orthodontic patients [52,53]. Therefore, more in-depth research is recommended in the future.

### Clinical Strengths, Limitations, and Future Research

Article 10 of the UN International Covenant on Civil and Political Rights stipulates that ‘all persons deprived of their liberty shall be treated with humanity and with respect for the inherent dignity of the human person’ [54,55]. In addition, the ‘United Nations Standard Minimum Rules for the Treatment of Prisons (the Nelson Mandela Rules)’ clearly state that prisoners, as citizens, should receive the same level of healthcare services as people in the general population [56]. Therefore, it is necessary to provide appropriate dental services as required by inmates and encourage them to practice self-administered oral health care through regular oral health education.

The correctional healthcare delivery system and health management in Korea are critical to the right to health of inmates and the right to healthcare in correctional facilities [1,57]. Thus, a comprehensive correctional healthcare delivery system, emphasising primary healthcare but providing other forms of care as well, is needed [1,57,58]. By emphasizing primary healthcare, the direction should be set to prevent the progression of mild diseases that can be cured in correctional facilities to a level requiring secondary or tertiary medical care [57].

This study analysed the factors related to OHRQoL in adult male inmates in Korea, which have not been reported on previously. Although the toothbrushing practice rate among inmates was high at 83.2%, it should be noted that OHRQoL was low, which was significantly associated with subjective oral symptoms, such as periodontal disease, malodour, chewing discomfort, temporomandibular joint disorder, and toothache. Thus, oral health education is necessary to enable proper oral health care among inmates according to their oral conditions.

The results of this study may facilitate the provision of dental care services to inmates and can help to improve the oral healthcare system. In particular, chewing discomfort and toothache among the perceived oral symptoms of male inmates were found to be important factors influencing OHRQoL. This finding may help to prioritize the dental care services needed for male inmates.

The limitations of this study were as follows: First, since this study was performed only with a self-reported questionnaire, without an oral examination, the relationship between OHRQoL and objective indicators of the oral health status could not be identified. Further studies need to perform oral examinations simultaneously to identify objective oral health indicators, such as dental caries experience, periodontal health, and chewing discomfort. Second, since only male inmates were selected as participants, we were unable to compare the results to female inmates. Further studies on female inmates are required to compare sex differences. Third, this study only targeted two prisons in the Republic of Korea and, thus, selection bias may have occurred. In the future, research with external validation should be conducted through a large-scale longitudinal study that expands the number of participants and identifies changes in OHRQoL according to various sociodemographic characteristics and the oral health status of inmates. Despite these limitations, the results derived herein are believed to contribute to the identification of appropriate oral health care methods and the scope of use of dental services that must be provided to inmates in each prison [59,60,61].

## 5. Conclusions

In this study, the practice rate of toothbrushing among male prison inmates in Korea was high at 83.2%; however, OHRQoL was somewhat low at 39.9 points. In addition, food impaction, sensitivity to cold/hot food, periodontal health, malodour, dry mouth, trouble biting/chewing, TMJ disorder, and toothache, which are perceived oral symptoms, were found to be positively correlated with each other, indicating the need for an integrated management of oral healthcare. In particular, factors affecting the OHIP-14 score were strongly negatively related to chewing discomfort and toothache. There was no significant relationship between self-esteem and the OHIP-14, and more in-depth research is required in the future. To guarantee the right to health among inmates in the future, our findings suggest the need for the establishment of a healthcare system that can provide appropriate dental care services and operate an oral health education programme to promote appropriate oral health awareness.

## Figures and Tables

**Figure 1 healthcare-11-02848-f001:**
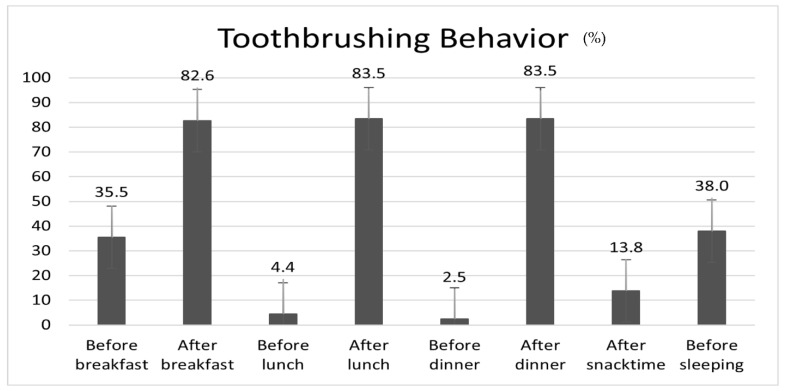
Toothbrushing behaviour in a day among adult male inmates.

**Figure 2 healthcare-11-02848-f002:**
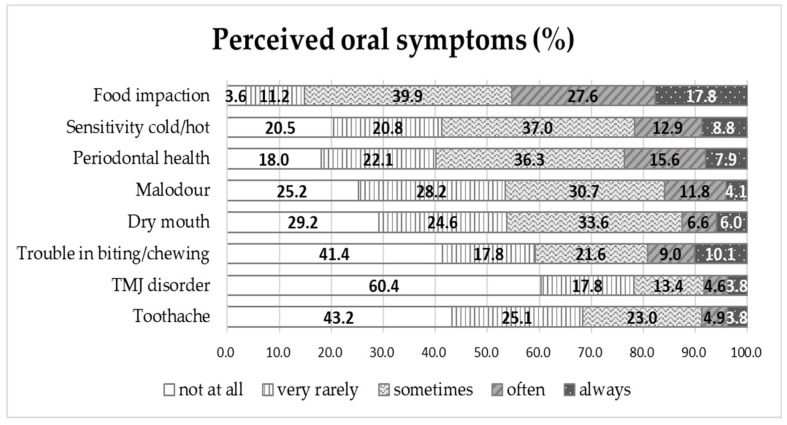
Perceived oral symptoms among adult male inmates; TMJ, temporomandibular joint.

**Table 1 healthcare-11-02848-t001:** The OHP-14 scores according to the generalized characteristics in adult male inmates.

Variables	Division	N (%)	OHIP-14	
			M ± SD	*p*-Value *
Age (years)	20–29 ^a^	37 (10.2)	46.86 ± 10.57	< 0.001
	30–39 ^ab^	80 (22.0)	43.60 ± 10.84	
	40–49 ^bc^	85 (23.4)	39.84 ± 13.39	
	50–59 ^c^	99 (27.3)	36.15 ± 12.07	
	≥ 60 ^c^	62 (17.1)	36.39 ± 11.28	
EDT (for last year)	Yes	110 (30.3)	38.26 ± 13.39	0.265
	No	246 (67.8)	40.59 ± 12.06	
	Do not know	7 (1.9)	39.71 ± 4.79	
EDE (for last year)	Yes	145 (39.8)	41.19 ± 12.04	0.247
	No	214 (59.0)	38.96 ± 12.70	
	Do not know	5 (1.4)	40.60 ± 5.59	
Toothbrushing time (min)	<1 ^a^	51 (14.01)	37.12 ± 12.37	0.020
	1 to less than 2 ^ab^	126 (34.62)	38.10 ± 13.06	
	2 to less than 3 ^b^	121 (33.24)	41.87 ± 10.84	
	≥3 ^b^	66 (18.13)	41.73 ± 13.12	
Use of fluoride toothpaste	Use	173 (47.92)	39.90 ± 11.89	0.432
	Do not use	73 (20.22)	38.60 ± 11.67	
	Do not know	115 (31.86)	40.97 ± 13.15	
Chewing discomfort	Tofu and Rice ^a^	21 (5.8)	25.81 ± 13.04	<0.001
	Apple ^a^	9 (2.5)	26.22 ± 15.47	
	Kimchi ^ab^	36 (10.0)	29.06 ± 12.80	
	Meat ^bc^	61 (16.9)	37.30 ± 90.4	
	Dried squid ^bc^	29 (8.1)	36.45 ± 11.53	
	Hard candy ^c^	204 (56.7)	45.46 ± 8.80	
Subjective health	Strongly disagree ^a^	56 (15.38)	26.41 ± 13.86	<0.001
	Disagree ^b^	93 (25.55)	36.47 ± 10.17	
	Neutral ^bc^	123 (33.79)	43.22 ± 8.95	
	Agree ^c^	82 (22.53)	46.89 ± 9.38	
	Strongly agree ^c^	10 (2.75)	48.10 ± 12.77	
Subjective oral health	Strongly disagree ^a^	78 (21.49)	27.77 ± 13.58	<0.001
	Disagree ^b^	130 (35.81)	38.68 ± 9.78	
	Neutral ^bc^	117 (32.23)	46.01 ± 8.00	
	Agree ^bc^	34 (9.37)	49.26 ± 7.57	
	Strongly agree ^c^	4 (1.10)	54.75 ± 1.89	
Self-esteem	High	49 (13.39)	41.10 ± 15.64	0.547
	Usual	252 (68.85)	40.12 ± 11.97	
	Low	65 (17.76)	38.65 ± 10.55	
Total		376 (100.0)	39.90 ± 12.38	

* Using one way ANOVA test at α = 0.05; ^a,b,c^ means followed with different letters are statistically significantly different according to Scheffe’s test at α = 0.05; EDT—experience of dental treatment; EDE—experience of dental examination.

**Table 2 healthcare-11-02848-t002:** Correlation between main variables and the OHIP-14 in adult male inmates.

	1	2	3	4	5	6	7	8	9	10
1. Food impaction	1									
2. Sensitivity to cold/hot	0.492 **	1								
3. Periodontal health	0.434 **	0.566 **	1							
4. Malodour	0.373 **	0.507 **	0.633 **	1						
5. Dry mouth	0.316 **	0.446 **	0.499 **	0.622 **	1					
6. Trouble biting/chewing	0.366 **	0.570 **	0.555 **	0.520 **	0.514 **	1				
7. TMJ disorder	0.176 **	0.340 **	0.386 **	0.431 **	0.426 **	0.447 **	1			
8. Toothache	0.318 **	0.509 **	0.548 **	0.536 **	0.520 **	0.628 **	0.583 **	1		
9. Self-esteem	0.018	0.008	−0.050	−0.114 *	−0.063	−0.040	−0037	−0.023	1	
10. OHIP-14	−0.338 **	−0.546 **	−0.570 **	−0.552 **	−0.523 **	−0.749 **	−0.488 **	−0.649 **	0.057	1

* *p* < 0.05; ** *p* < 0.01 using Pearson’s correlation coefficient at α = 0.01; OHIP-14—oral health impact profile-14; TMJ—temporomandibular joint.

**Table 3 healthcare-11-02848-t003:** Factors related to the OHIP-14 in adult male inmates.

	Model 1					Model 2				
	B	SE	ß	t	*p*-Value *	B	SE	ß	t	*p*-Value *
(Constant)	20.705	2.998		6.905	<0.001	53.394	3.381		15.790	<0.001
Age	−0.093	0.038	−0.099	−2.436	0.015	−0.138	0.031	−0.146	−4.412	<0.001
Subjective health	2.259	0.597	0.200	3.785	<0.001	1.482	0.471	0.131	3.146	0.002
Subjective oral health	3.493	0.676	0.278	5.168	<0.001	0.755	0.570	0.060	1.326	0.186
Chewing discomfort (ref. = tofu and rice)										
Apple	−2.735	3.569	−0.034	−0.766	0.444	−4.808	2.809	−0.060	−1.712	0.088
Kimchi	3.100	2.400	0.077	1.292	0.197	0.273	1.908	0.007	0.143	0.886
Meat	8.629	2.207	0.270	3.910	<0.001	3.196	1.778	0.100	1.798	0.073
Squid	5.675	2.548	0.126	2.228	0.027	1.279	2.041	0.028	0.627	0.531
Candy	12.665	2.080	0.522	6.089	<0.001	4.255	1.768	0.175	2.407	0.017
Perceived oral symptoms										
Food impaction						0.212	0.423	0.018	0.502	0.616
Sensitivity to cold/hot						0.400	0.429	0.039	0.931	0.353
Periodontal health						−0.936	0.463	−0.089	−2.021	0.044
Malodour						−1.423	0.481	−0.131	−2.957	0.003
Dry mouth						−0.008	0.425	−0.001	−0.018	0.985
Trouble biting/chewing						−2.735	0.429	−0.307	−6.376	<0.001
TMJ disorder						−1.086	0.409	−0.099	−2.656	0.008
Toothache						−1.687	0.491	−0.154	−3.439	0.001
*F* (*p*-Value)	44.754 (<0.001)					51.631 (<0.001)				
*R* ^2^	0.511					0.711				
Adj. *R*^2^	0.499					0.698				

* Using hierarchical multiple linear regression at α = 0.05; TMJ—temporomandibular joint.

## Data Availability

The data presented in this study are available on reasonable request from the corresponding author.

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
