# Peer review of "Factors Related to Oral-Health-Related Quality of Life in Adult Male Inmates: A Cross-Sectional Study"

_healthcare, 2023, doi:10.3390/healthcare11212848_

Round 1

Reviewer 1 Report

Comments and Suggestions for Authors

First of all, I would like to greet the authors and congratulate them on the theme and work done. The study appears correctly performed and written without logical or factual errors.

Authors have well revised the main issue in this cross-sectional study. The following comments are addressed and require minor modifications to enhance the quality of the manuscript:

Introduction is clearly disclosed and supported by actual references.Do you consider your study original or relevant in the field? Does it address a specific gap in the field? What does it add to the area, compared with other published material?

2.The methodology is very clear and without apparent errors, my only doubt concerns the inclusion criteria, could the authors inform about this? Another aspect I suggest is to provide, in the manuscript,  the questionnaire sheet used.

3.The results seem correctly presented, being objective but exhaustive and well explained through the tables presented.

4.About discussion I think it is well structured contrasting with some previously published studies. I would like to see clarified if  oral health care  is provided  in prisons in Korea or some protocol with educational institutions or private clinics. If not, this would be a great suggestion to minimize the problem.

I think the major limitation about this study, as well pointed by the authors, is the lack of oral examination to give greater robustness and evidence to the correlations studied. That been said, I found

5.Conclusions objective consistent with the evidence and arguments presented  and responding to the research question.

Author Response

First of all, I would like to greet the authors and congratulate them on the theme and work done. The study appears correctly performed and written without logical or factual errors.

Authors have well revised the main issue in this cross-sectional study. The following comments are addressed and require minor modifications to enhance the quality of the manuscript:

  1. Introduction is clearly disclosed and supported by actual references. Do you consider your study original or relevant in the field? Does it address a specific gap in the field? What does it add to the area, compared with other published material?

Authors’ response:

Thank you for your thoughtful comments. In Korea, there is very little research on oral health among prison inmates. In particular, it is difficult to find studies exploring factors related to the oral health-related quality of life in terms of oral health behavior and oral self-report symptoms. As cited in the introduction, previous studies have reported that the prevalence of dental caries among prison inmates is lower than that among the general population, but there is a lack of research identifying influencing factors related to oral health. Therefore, based on previous research, the results of this study will contribute to determining the priorities of oral-health promotion programs and dental services needed by prisoners by identifying important influencing factors on the oral health of prison inmates in Korea. In addition, it is necessary to operate an oral health education program so that the rate of proper oral health practice among prison inmates can be increased. This content was also presented as the significance of the research in the review, but some of the content has been revised as follows:

  • Although the toothbrushing practice rate among inmates was high at 83.2%, it should be noted that OHRQoL was low, which was significantly associated with subjective oral symptoms such as periodontal disease, malodour, chewing discomfort, temporomandibular joint disorder, and toothache. Thus, oral health education is necessary to enable proper oral health care among inmates according to their oral condition.
  1. The methodology is very clear and without apparent errors, my only doubt concerns the inclusion criteria, could the authors inform about this? Another aspect I suggest is to provide, in the manuscript, the questionnaire sheet used.

Authors’ response:

Thank you for your constructive comments. The selection criteria for participants in this study were those currently incarcerated in prison, as indicated in section “2.2. Participants,” which was verified during the IRB approval process. The data were registered with the Korea Clinical Research Information Service (KCT 0008129). Furthermore, section “2.4. Data Collection” describes the data collection procedure in terms of the ethical consideration of participants who met the selection criteria. In addition, the text describing the selection and exclusion criteria for participants has been revised as follows:

  • The participants selected were inmates who voluntarily agreed to respond to the questionnaire. We excluded those who withdrew from participation during the course of the survey as they felt too uncomfortable with the questions asked to proceed further, as well as those who were unable to participate due to quarantine or punishment.
  1. The results seem correctly presented, being objective but exhaustive and well explained through the tables presented.

Authors’ response:

Thank you for your helpful comments. The manuscript has been reviewed overall, and some errors were revised.

  1. About discussion I think it is well structured contrasting with some previously published studies. I would like to see clarified if oral health care is provided  in prisons in Korea or some protocol with educational institutions or private clinics. If not, this would be a great suggestion to minimize the problem.

I think the major limitation about this study, as well pointed by the authors, is the lack of oral examination to give greater robustness and evidence to the correlations studied. That been said, I found

Authors’ response:

Thank you for your thoughtful comments. In Korea, regular oral examinations are performed on prison inmates at the national level, but unfortunately, the variables covered in our study are not investigated. Our study is the first survey in Korea to investigate oral symptoms and oral health-related quality of life. However, because oral examinations could not be performed on participants simultaneously, the relationship between objective oral indicators and subjective oral health levels could not be determined. Therefore, future research will need to conduct oral examinations and identify objective oral indicators simultaneously. These points are covered in the discussion.

  1. Conclusions objective consistent with the evidence and arguments presented and responding to the research question.

Authors’ response:

Accordingly, we have revised the conclusions as follows:

  • There was no significant relationship between self-esteem and OHIP-14, and more in-depth research is required in the future.

Reviewer 2 Report

Comments and Suggestions for Authors

I was pleased to read the manuscript entitled "Factors Related to Oral Health-Related Quality of Life in Adult Male Inmates: A Cross-sectional Study" and to review it.

The study investigated oral health behaviours and oral symptoms among male inmates and analysed factors related to oral health related quality of life (OHRQoL) The article is written in a typical format and is well structured, but some corrections may be appropriate.

Title – The title is informative and it accurately reflect the content of the manuscript.

Abstract – The abstract is more or less complete and adequately reflects the content of the manuscript. However, I would recommend to revise the first sentence "The health of inmates living in a restricted environment are affected by oral health behaviours and is related to their oral health-related quality of life (OHRQoL)." This sentence is wrong about what is cause and what is effect.

Introduction – the Introduction provide sufficient theoretical background for the study. All examined research questions and/or hypotheses were introduced The introduction is structured logically and the text is fluent. The rationale of the study is well described and the study problem is stated clearly. Relevant and unbiased literature was used, however, references from #4 to #9 are not much specific. Instead, I suggest selecting review articles and monograph-level publications.

Method – Sampling and measures are shortly described and are appropriate to answer the proposed research questions. Here I recommend:

- In the sample size calculation description (lines 101-105), specify which 'test family' and 'statistical test' were selected when the calculations were performed with G*Power.

- Rosenberg's Self-esteem scale has 10 items. Is it correct that you indicated that this scale has 11 items (see line 142)?

Results – In general, results are clearly organized and presented. However, several inaccuracies were noted:

Discussion – the structure of the Discussion is clear. The interpretations is appropriate and is supported by the results. The study findings are discussed with relevant literature. However, the contribution of the study to the field is weakly explained - it is recommended to emphasize the novelty of the study.

Conclusions – are supported with the study results.

Thank you for considering my opinion. I encourage authors to keep on working to improve the manuscript.

Comments on the Quality of English Language

Minor editing of English language required

Author Response

I was pleased to read the manuscript entitled "Factors Related to Oral Health-Related Quality of Life in Adult Male Inmates: A Cross-sectional Study" and to review it.

The study investigated oral health behaviours and oral symptoms among male inmates and analysed factors related to oral health related quality of life (OHRQoL) The article is written in a typical format and is well structured, but some corrections may be appropriate.

  1. Title – The title is informative and it accurately reflect the content of the manuscript.

Abstract – The abstract is more or less complete and adequately reflects the content of the manuscript. However, I would recommend to revise the first sentence "The health of inmates living in a restricted environment are affected by oral health behaviours and is related to their oral health-related quality of life (OHRQoL)." This sentence is wrong about what is cause and what is effect.

Authors’ response:

Thank you for your constructive comments. We have revised the text accordingly:

  • The level of oral health among inmates living in restricted environments is poor.
  1. Introduction – the Introduction provide sufficient theoretical background for the study. All examined research questions and/or hypotheses were introduced The introduction is structured logically and the text is fluent. The rationale of the study is well described and the study problem is stated clearly. Relevant and unbiased literature was used, however, references from #4 to #9 are not much specific. Instead, I suggest selecting review articles and monograph-level publications.

Authors’ response:

Thank you for your valuable comments. References #4 to #9 have been replaced with #4 to #7 in the revised manuscript as follows:

  1. Sharma, N.; Bhatia, S.; Sodhi, A. S.; Batra, N. Oral Microbiome and Health. AIMS Microbiol.20184 (1), 42–66. DOI: 10.3934/microbiol.2018.1.42
  2. Kozak, M.; Pawlik, A. The Role of the Oral Microbiome in the Development of Diseases. Int. J. Mol. Sci.202324 (6), 5231. DOI: 10.3390/ijms24065231
  3. Guzman, C.; Zaclli, A.; Molinari, J. Streptococcus cristatusBacteremia in a Patient with Poor Oral Hygiene: A Case Report. J. Med. Case Rep. 202317 (1), 218. DOI: 10.1186/s13256-023-03818-z
  4. Amato, A. Periodontitis and Cancer: Beyond the Boundaries of Oral Cavity. Cancers202315 (6), 1736. DOI: 10.3390/cancers15061736

  1. Method – Sampling and measures are shortly described and are appropriate to answer the proposed research questions. Here I recommend:

- In the sample size calculation description (lines 101-105), specify which 'test family' and 'statistical test' were selected when the calculations were performed with G*Power.

Authors’ response:

Thank you for your constructive comments. In our study, the sample-size calculation was performed using multiple linear regression analysis with 27 independent and covariate variables as predictors. The text on the research method has been revised as follows:

  • With the effect size set at 0.10, power at 0.90, and 27 predictive factors using multiple regression analysis, we calculated that 313 participants were required.

- Rosenberg's Self-esteem scale has 10 items. Is it correct that you indicated that this scale has 11 items (see line 142)?

Authors’ response:

Thank you for your thoughtful comments. The number of questions present in Rosenberg's self-esteem scale has been revised to 10.

  1. Results – In general, results are clearly organized and presented. However, several inaccuracies were noted:

Authors’ response:

Thank you for your thoughtful comments. To improve the readability of our paper, standard deviation was added to the mean of OHIP-14 scores based on general characteristics. In addition, the mean error was revised in the 20s (46.86 ± 10.57).

  1. Discussion – the structure of the Discussion is clear. The interpretations is appropriate and is supported by the results. The study findings are discussed with relevant literature. However, the contribution of the study to the field is weakly explained - it is recommended to emphasize the novelty of the study.

Conclusions – are supported with the study results.

Thank you for considering my opinion. I encourage authors to keep on working to improve the manuscript.

Authors’ response:

Thank you for your constructive comments. Accordingly, we have added the following text to the discussion to further highlight the significance of the research results:

  • Although the toothbrushing practice rate among inmates was high at 83.2%, it should be noted that OHRQoL was low, which was significantly associated with subjective oral symptoms such as periodontal disease, malodour, chewing discomfort, temporoman-dibular joint disorder, and toothache. Thus, oral health education is necessary to enable proper oral health care among inmates according to their oral condition.

Reviewer 3 Report

Comments and Suggestions for Authors

Review

Manuscript details:  Journal: Healthcare
Manuscript ID: healthcare-2624329
Type of manuscript: Article
Title: Factors Related to Oral Health-Related Quality of Life in Adult Male
Inmates: A Cross-sectional Study  
Authors: Hae-Soo Yoon, Kyoung-Sun Kim, Jong-Hwa Jang *

The aim of this study is to investigate oral health, perceived oral symptoms, and OHRQoL levels among Korean prison inmates. Next, the authors identify relationships and factors between symptoms affecting OHRQoL which could serve as a basis for the development of programs aimed at improving the oral health of prisoners.

The subjective nature of the study undermines the validity of the results. An in-depth individual general and oral examination (caries, periodontal diseases, wear, abrasion, etc. quality of care) would provide some insight into the reality of the oral dental condition of prisoners in Korea. The sample of the population studied presents a disparity regarding their ages. By targeting a specific age group this can refine the results. More details on the absence of teeth or the presence of a removable or fixed prosthesis would also be useful to enhance the value of the article.

Lines 47-48 : avoid repeating information already stated lines 42-43.

Line 55 : avoid repeating distability.

Line 93 : Why did you confine your investigation to men only?

Line 108-109 : Could these conditions constitute a bias that could modify the final result?

Lien 329: This ability to chew is directly related to the number of teeth present and to the presence or absence of a removable prosthesis. this needs to be clarified.

Line 342: which shows the subjective nature of the study!

Line 350 (paragraph): There is such a disparity between countries at this level that this remains wishful thinking for the moment.

Line 371 ( paragraph): At this level you can mention the desirable management of oral dental care either by the various protection organizations depending on the states or access to care for the prison population provided by private dental surgeons.

Line 383: The subjective nature of the study on this parameter requires reservations.

Bibliography

Bibliography providing information on care intended for prison population. Hospitalized inmates (Briggs) tele expertise (Novais, Inquimbert).

Briggs MS, Kolbus ES, Patterson KM, Harmon-Matthews LE, McGrath S, Quatman-Yates CC, Meirelles C, Salsberry MJ. Role of Oral Intake, Mobility, and Activity Measures in Informing Discharge Recommendations for Hospitalized Inmate and Noninmate Patients With COVID-19: Retrospective Analysis. JMIR Rehabil Assist Technol. 2023 Jun 27;10:e43250. doi: 10.2196/43250. PMID: 37224276; PMCID: PMC10337323.

Novais A, Fac C, Allouche M, Atallah É, Godkine N, Guyader T, Hariga A, Hoarau W, Oulmi A, Trumbic F, Goujard C, Pirnay P. [Télédent, an oral tele-expertise experience in a penitentiary environment]. Med Sci (Paris). 2019 Nov;35(11):866-870. French. doi: 10.1051/medsci/2019168. Epub 2019 Dec 17. PMID: 31845878.

Inquimbert C, Balacianu I, Huyghe N, Pasdeloup J, Tramini P, Meroueh F, Montal S, Bencharit S, Giraudeau N. Applications of teledentistry in a French inmate population: A one-year observational study. PLoS One. 2021 Apr 7;16(4):e0247778. doi: 10.1371/journal.pone.0247778. PMID: 33826659; PMCID: PMC8026055.

Author Response

The aim of this study is to investigate oral health, perceived oral symptoms, and OHRQoL levels among Korean prison inmates. Next, the authors identify relationships and factors between symptoms affecting OHRQoL which could serve as a basis for the development of programs aimed at improving the oral health of prisoners.

The subjective nature of the study undermines the validity of the results. An in-depth individual general and oral examination (caries, periodontal diseases, wear, abrasion, etc. quality of care) would provide some insight into the reality of the oral dental condition of prisoners in Korea. The sample of the population studied presents a disparity regarding their ages. By targeting a specific age group this can refine the results. More details on the absence of teeth or the presence of a removable or fixed prosthesis would also be useful to enhance the value of the article.

Authors’ response:

Thank you for your constructive comments. As suggested as a limitation of the study, our study results were based on a survey, and thus, oral examinations were not conducted. However, we believe that it is meaningful that we explored related factors that affect the oral health of prisoners by investigating oral health behavior, oral self-awareness symptoms, and oral health-related quality of life. As a result of the actual analysis, prisoners had a high rate of brushing their teeth, but their quality of life related to oral health was low; this observation was related to oral self-awareness symptoms. As you have indicated in your comment, we believe it is important to determine the presence or absence of a removable prosthesis through oral examination along with a survey in future research. Accordingly, future research directions were suggested in the discussion section, and the clinical significance of this study was added as follows:

  • Although the toothbrushing practice rate among inmates was high at 83.2%, it should be noted that OHRQoL was low, which was significantly associated with subjective oral symptoms such as periodontal disease, malodour, chewing discomfort, temporoman-dibular joint disorder, and toothache. Thus, oral health education is necessary to enable proper oral health care among inmates according to their oral condition.

Lines 47-48 : avoid repeating information already stated lines 42-43.

Authors’ response:

Thank you for your thoughtful comments. As recommended, the following text has been deleted.

  • Moreover, oral health is strongly associated with systemic health, and oral diseases are associated with major systemic diseases, including hypertension and diabetes [4-6,8].

Line 55 : avoid repeating distability.

Authors’ response:

Thank you for your meaningful comments. The text has been revised as follows:

  • This questionnaire comprises concepts such as functional limitation, physical pain, psychological discomfort, physical disability, and social handicaps.

Line 93 : Why did you confine your investigation to men only?

Authors’ response:

In Korea, male and female inmates are housed in separate prisons. For our study, inmates in two men's prisons were selected as subjects. In the discussion section, future research was suggested to be conducted to compare sex differences by studying female prison inmates as follows:

  • Second, since only male inmates were selected as participants, we were unable to compare the results in female inmates. Further studies on female inmates are required to compare sex differences.

Line 108-109 : Could these conditions constitute a bias that could modify the final result?

Authors’ response:

Our study participants were inmates who were able to respond during the survey period, and there were no measures to exclude those who could not participate in the survey owing to actual quarantine or punishment; thus, the following sentence was deleted.

  • We excluded those who withdrew from participation during the course of the survey as they felt too uncomfortable with the questions asked to proceed further, as well as those who were unable to participate due to quarantine or punishment.

Lien 329: This ability to chew is directly related to the number of teeth present and to the presence or absence of a removable prosthesis. this needs to be clarified.

Authors’ response:

We agree with your constructive comments and have added a paragraph as follows:

  • These results suggest that better chewing ability indicates higher OHRQoL. In future studies, it is necessary to investigate the number of teeth in inmates and the use of dental prosthetics to determine relationships with clinical indicators.

Line 342: which shows the subjective nature of the study!

Authors’ response:

Although no significant relationship was found in our study, the results showing that higher self-esteem was associated with higher oral health-related quality of life are similar to those reported in studies by several groups. For clarity, the following text has been added:

  • Therefore, more in-depth research is suggested in the future.

Line 350 (paragraph): There is such a disparity between countries at this level that this remains wishful thinking for the moment.

Authors’ response:

Thank you for your valuable comments. We have added the following text to this paragraph:

  • Therefore, it is necessary to provide appropriate dental services as required by inmates and encourage them to practice self-administered oral health care through regular oral-health education.

Line 371 ( paragraph): At this level you can mention the desirable management of oral dental care either by the various protection organizations depending on the states or access to care for the prison population provided by private dental surgeons.

Authors’ response:

Accordingly, we have revised the paragraph as follows and cited related literature (references #61 to #63):

  • Third, this study only targeted two prisons in the Republic of Korea, and thus, selection bias may occur. In the future, research with external validation should be conducted through a large-scale longitudinal study that can expand the number of participants and identify changes in OHRQoL according to various socio-demographic characteristics and oral health status of inmates. Despite these limitations, the results derived herein are believed to contribute to the identification of appropriate oral health care methods and the scope of use of dental services that must be provides to inmates in each prison [61-63].
  • Briggs, M. S.; Kolbus, E. S.; Patterson, K. M.; Harmon-Matthews, L. E.; McGrath, S.; Quatman-Yates, C. C.; Meirelles, C.; Salsberry, M. J. Role of Oral Intake, Mobility, and Activity Measures in Informing Discharge Recommendations for Hospitalized Inmate and Noninmate Patients with COVID-19: Retrospective Analysis. J.M.I.R. Rehabil. Assist. Technol.2023, 10, e43250. DOI: 10.2196/43250
  • Novais, A.; Fac, C.; Allouche, M.; Atallah, É.; Godkine, N.; Guyader, T.; Hariga, A.; Hoarau, W.; Oulmi, A.; Trumbic, F.; Goujard, C.; Pirnay, P. [Télédent, an Oral Tele-Expertise Experience in a Penitentiary Environment]. Med. Sci. (Paris)2019, 35 (11), 866–870. DOI: 10.1051/medsci/2019168
  • Inquimbert, C.; Balacianu, I.; Huyghe, N.; Pasdeloup, J.; Tramini, P.; Meroueh, F.; Montal, S.; Bencharit, S.; Giraudeau, N. Applications of Teledentistry in a French Inmate Population: A One-Year Observational Study. PLOS ONE2021, 16 (4), e0247778. DOI: 10.1371/journal.pone.0247778

Line 383: The subjective nature of the study on this parameter requires reservations.

Authors’ response:

Accordingly, we have revised the text as follows:

  • There was no significant relationship between self-esteem and OHIP-14, and more in-depth research is required in the future.

Reviewer 4 Report

Comments and Suggestions for Authors

Dear authors

First, I would like to congratulate you on your interesting and well-written cross-sectional study about factors related to the oral health-related quality of life (OHRQoL) in 376 adult male inmates from two Korean prisons, who provided insights by filling out questionnaires. I identified the following key messages in your manuscript:

1.       Although over 80 % of participants brushed their teeth a after lunch and dinner, the OHRQoL was low.

2.       While trouble with biting/chewing, toothache, temporomandibular joint disorder, and periodontal health were identified as factors related to OHRQoL, self-esteem was not significantly correlated to it.

3.       The results of the study highlight the need for oral health education and dental care programs for inmates to support both, their oral health and their OHRQoL.

The following comments could be considered during revision:

1.       Results

-          How was the response rate of inmates to your questionnaire? Please consider adding this information to your manuscript.

-          Page 5, Figure 2: It seems like data for “food impaction” is cut off by the percentage symbol. I advise you to check this.

-          Page 7, line 203: I think there is a small typing error in the following sentence: “(…) OHIP-14 score was higher for those in their 20s (46.89) (…)” -> In your table it is 46.86. - please check this.

-          Please consider adding the standard deviation to the mean values presented in the text, as this allows the reader to see the amount variation in the values while reading the text.

2.       Discussion

-          Page 10, line 282-283: I think there is a small typing error in the following sentence: “(…) the practice rate before sleeping (38.8%) was rather low (…)”. According to Figure 1, it is 38.0 %. Please check this number.

-          Do you have any information as to whether the duration of being imprisoned or the socio-economic status before imprisoning influences inmates’ OHRQoL?

Thank you in advance for considering my comments during revision. Good luck and kind regards!

Comments on the Quality of English Language

The manuscript is well written; however, it could benefit from some minor editing of English language.

Author Response

Dear authors

First, I would like to congratulate you on your interesting and well-written cross-sectional study about factors related to the oral health-related quality of life (OHRQoL) in 376 adult male inmates from two Korean prisons, who provided insights by filling out questionnaires. I identified the following key messages in your manuscript:

  1. Although over 80 % of participants brushed their teeth a after lunch and dinner, the OHRQoL was low.
  2. While trouble with biting/chewing, toothache, temporomandibular joint disorder, and periodontal health were identified as factors related to OHRQoL, self-esteem was not significantly correlated to it.
  3. The results of the study highlight the need for oral health education and dental care programs for inmates to support both, their oral health and their OHRQoL.

The following comments could be considered during revision:

  1. Results- How was the response rate of inmates to your questionnaire? Please consider adding this information to your manuscript.

Authors’ response:

Thank you for your constructive comments. In the research methods section, “2.4. Data Collection”, the response rate (%) of the number of participants has been added as follows:

  • No incomplete questionnaires were included; thus, 376 questionnaires (100.0%) were included for analysis.

- Page 5, Figure 2  It seems like data for “food impaction” is cut off by the percentage symbol. I advise you to check this.

Authors’ response:

Thank you for your careful comments. Fig. 2 has been revised for readability.

- Page 7, line 203: I think there is a small typing error in the following sentence: “(…) OHIP-14 score was higher for those in their 20s (46.89) (…)” -> In your table it is 46.86. - please check this.

Authors’ response:

Thank you for your careful comments. The error has been revised as follows:

  • The OHIP-14 score was higher for those in their 20s (46.86 ± 10.57) than for those in their 40s (39.84 ± 13.39), 50s (36.15 ± 12.07), and 60s or older (36.39 ± 11.28), and the scores of those in their 30s (43.60 ± 10.84) was significantly different from that of those in their 40s or older.

- Please consider adding the standard deviation to the mean values presented in the text, as this allows the reader to see the amount variation in the values while reading the text.

Authors’ response:

We agree with your constructive comments and have added the standard deviation to the mean.

  1. Discussion

- Page 10, line 282-283: I think there is a small typing error in the following sentence: “(…) the practice rate before sleeping (38.8%) was rather low (…)”. According to Figure 1, it is 38.0 %. Please check this number.

Authors’ response:

Thank you for your careful comments. The error has been revised as follows:

  • Although the mean practice rate of toothbrushing after meals among male inmates in Korea was 83.2%, the practice rate before breakfast (35.5%) was higher, and the practice rate before sleeping (0%) was rather low.

- Do you have any information as to whether the duration of being imprisoned or the socio-economic status before imprisoning influences inmates’ OHRQoL?

Thank you in advance for considering my comments during revision. Good luck and kind regards!

Authors’ response:

Thank you for your constructive comments. Your suggestions have been added to the limitations section as future research directions, as follows:

  • Third, this study only targeted two prisons in the Republic of Korea, and thus, selection bias may occur. In the future, research with external validation should be conducted through a large-scale longitudinal study that can expand the number of participants and identify changes in OHRQoL according to various socio-demographic characteristics and oral health status of inmates. Despite these limitations, the results derived herein are believed to contribute to the identification of appropriate oral health care methods and the scope of use of dental services that must be provides to inmates in each prison [61-63].

Round 2

Reviewer 3 Report

Comments and Suggestions for Authors

The requested modifications having been made and the reservations concerning the results there is no disadvantage in publishing this article.